# Unveiling the Neuroprotective Potential of Date Palm (*Phoenix dactylifera*): A Systematic Review

**DOI:** 10.3390/ph17091221

**Published:** 2024-09-17

**Authors:** Syed Mohammed Basheeruddin Asdaq, Abdulaziz Ali Almutiri, Abdullah Alenzi, Maheen Shaikh, Mujeeb Ahmed Shaik, Sultan Alshehri, Syed Imam Rabbani

**Affiliations:** 1Department of Pharmacy Practice, College of Pharmacy, AlMaarefa University, Dariyah, Riyadh 13713, Saudi Arabia; 191120097@student.um.edu.sa (A.A.A.); 202120443@student.um.edu.sa (A.A.); 2College of Medicine, Alfaisal University, Riyadh 11533, Saudi Arabia; maheensskh@gmail.com; 3Department of Basic Medical Science, College of Medicine, AlMaarefa University, Dariyah, Riyadh 13713, Saudi Arabia; smujeeb@um.edu.sa; 4Department of Pharmaceutical Sciences, College of Pharmacy, AlMaarefa University, Dariyah, Riyadh 13713, Saudi Arabia; sshehri@um.edu.sa; 5Department of Pharmacology and Toxicology, College of Pharmacy, Qassim University, Buraydah 51452, Saudi Arabia; s.rabbani@qu.edu.sa

**Keywords:** drug efficacy, *Pheonix dactylifera*, antioxidant, neuroprotection, oxidative stress, phenolic compounds

## Abstract

**Background:** Neurodegenerative diseases primarily afflict the elderly and are characterized by a progressive loss of neurons. Oxidative stress is intricately linked to the advancement of these conditions. This study focuses on *Phoenix dactylifera* (*P. dactylifera;* Family: *Arecaceae*), commonly known as “Ajwa,” a globally cultivated herbal plant renowned for its potent antioxidant properties and reported neuroprotective effects in pharmacological studies. **Method:** This comprehensive systematic review delves into the antioxidant properties of plant extracts and their phytochemical components, with a particular emphasis on *P. dactylifera* and its potential neuroprotective benefits. Preferred reporting items for systemic reviews and meta-analysis (PRISMA) were employed to review the articles. **Results:** The study includes 269 articles published in the literature and 17 were selected after qualitative analysis. The growing body of research underscores the critical role of polyphenolic compounds found in *P. dactylifera*, which significantly contribute to its neuroprotective effects through antioxidant mechanisms. Despite emerging insights into the antioxidant actions of *P. dactylifera*, further investigation is essential to fully elucidate the specific pathways through which it confers neuroprotection. **Conclusions:** Like many other plant-based supplements, *P. dactylifera*’s antioxidant effects are likely mediated by synergistic interactions among its diverse bioactive compounds, rather than by any single constituent alone. Therefore, additional preclinical and clinical studies are necessary to explore *P. dactylifera*’s therapeutic potential comprehensively, especially in terms of its targeted antioxidant activities aimed at mitigating neurodegenerative processes. Such research holds promise for advancing our understanding and potentially harnessing the therapeutic benefits of *P. dactylifera* in neuroprotection.

## 1. Introduction

The survival of living things depends heavily on oxygen, which also serves as the building block for cellular respiration, which results in aerobic respiration [1]. Numerous molecules and cellular structures can be harmed by oxidative stress (OS), which can alter how well organs and systems function. The body builds up OS through both internal and external mechanisms [2]. Chronic exposure to chemicals including medications may result in modifications to the host system functioning [3]. An overabundance of reactive oxygen species (ROS) can harm nucleic acids, membrane lipids, and cellular proteins, impairing normal cellular function [4,5]. Furthermore, NO^.^ radicals produced by nitric oxide have also been reported to induce damaging effects on organs such as the endothelium [6].

According to the literature, several conventional medications used in the treatment of several diseases are losing their efficacy [7]. Today, though, the focus is shifting away from synthetic medications and toward natural medications derived from bacteria or plants to treat illnesses [8]. The potential pharmacological value of natural products is continuously being investigated, with a focus on their potential effects on amoebicidal, cytotoxic, antimicrobial, spasmolytic, bronchodilator, antioxidant, anti-diarrheal, anti-Parkinsonism, anti-inflammatory, hypotensive, hepatoprotective, and hypoglycaemic functions [9].

The use of various herbs and medicinal plants as supplements to maintain overall mental well-being has drawn a lot of interest, especially considering their potential to improve memory and protect against neurodegeneration through antioxidant properties [10]. These plants have been investigated for likely therapeutic use in neurodegenerative disorders, with Alzheimer’s disease being the more prevalent type, followed by Parkinson’s disease [11]. Ginsenosides, the primary active plant element of various Panax species, have been the subject of intensive research due to their potential protective effects against neurological illnesses [12]. This has led to the practice of ginsenosides as a general tonic for enhancing well-being and managing stress [13]. Similarly, increasing scientific data showing that *Ginkgo biloba* supplements are effective in treating a variety of neurological disorders, such as ischemic stroke and cognitive impairment, has led to their classification as high-claim products [14].

The *Arecaceae* family, sometimes called the “Palm” family, comprises 2000 genera and 4000 species. Major crops from this family include dates, coconuts, and African palm oil [15]. Five of the 12 species in the Phoenix genus, including *Phoenix dactylifera*, are edible. Worldwide, there are roughly 3000 growers of the palm family [16]. The date palm is indigenous to the Persian Gulf and North Africa; still, its exact origin is unknown. Iraq, Egypt, Saudi Arabia, Tunisia, Algeria, the United Arab Emirates, Oman, Libya, Saudi Arabia, Pakistan, Sudan, Europe, and the United States are the top 10 producers of date palms [17].

*Phoenix dactylifera*, also known as Ajwa, is a variety of dates that is grown in different countries, including Saudi Arabia. It is useful in the treatment of many disorders and has been shown to have a protective effect against liver toxicity [18]. Regarding the potential health risks of dates, several biological and folkloric activities based on in vitro and animal models have been reported. These comprise protecting the heart and circulatory system by lowering blood levels of triglycerides and LDL cholesterol, preventing atherosclerosis and heart problems, preventing anemia, and maintaining a healthy nervous system and energy production [19,20,21,22]. Figure 1 represents *Phoenix dactylifera* cultivated in Saudi Arabia. Due to their high degree of nutraceutical potential for boosting the power of resistance, *P. dactylifera* dates and their preparations may be used in the care of long-suffering patients with considerably weakened or suppressed immune systems [23].

Furthermore, it has been found that cyanidin inhibits the death of neurons caused by amyloid beta by reducing reactive oxygen species (ROS) and reactive nitrogen species (RNS), which are linked to the modification of the mitochondrial death pathway in SK-N-SH cells [24,25]. To the best of our knowledge, a widespread published literature review has not previously been conducted on the protective effects of this plant product or on preparation against neurodegeneration brought on by oxidative stress. To investigate the possible advantages of *Pheonix dactylifera* against ROS-mediated neurodegeneration, a comprehensive assessment of the literature’s several scientific studies on the subject was planned for this study.

## 2. Methodology

This systematic review was prepared following Preferred Reporting Items for Systematic Reviews and Meta-Analyses (PRISMA) standards (registration number IRB24-007). This study conducted the systemic review using the procedures outlined in the literature [26].

### 2.1. Literature Search Strategy

From December 2023 to March 2024, electronic literature searches of PubMed (n = 216), SCOPUS (n = 26), Web of Science (n = 15), and BIOSIS (n = 12) were conducted using keywords like *Pheonix dactylifera* OR *P. dactylifera* OR Ajwa OR Ajwa dates AND antioxidant OR oxidative stress OR free radicals OR reactive oxygen species AND neuroprotection OR neurodegeneration (Appendix A).

### 2.2. Eligibility Criteria

Studies satisfying the following criteria were included in this review:Full-length, English-language articles that include detailed information, particularly details about the *P. dactylifera* research.Cross-sectional studies that were released in the previous two years, 2022–2023.Investigations that were conducted to ascertain the impact of *P. dactylifera* and oxidative stress on neurodegenerative disorders.Information about *P. dactylifera*’s pharmacological properties, including dose, duration, side reactions, and potential mechanisms of action.Investigated and evaluated publications from indexed journals that provide comprehensive details on statistics and their significance level.

Studies that failed to meet the eligibility criteria were excluded. Furthermore, data that overlapped or was duplicated, as well as information that could not be recovered, were not included in the analysis [27].

### 2.3. Study Selection

Two authors of this work independently reviewed the literature to determine *P. dactylifera*’s involvement in neurodegenerative illnesses. To decrease the chance of overlapping data, the authors segregated the diseases (e.g., Alzheimer’s’ disease, Parkinson’s disease, Huntington’s disease, amyotrophic lateral sclerosis, Lewy body disease, and spinal muscular atrophy) and analyzed the role of the plant on individual diseases. There were mainly two steps in the eligibility screening procedure. The author started by going over the titles and abstracts of the documents that were obtained. A thorough screening of the full-text articles chosen in the first phase was carried out in the second step. Any differences that emerged from the results were addressed and settled by the other authors of the study [28].

### 2.4. Data Extraction

Relevant data were separately retrieved using a pre-structured data extraction form. The retrieved data contained several crucial components, such as the study design, the parameters that were measured, and specifics on significant discoveries. Other elements that were crucial for evaluating the articles’ content included the language used in the publication, study design, strain and quantity of animals, protocol, dosage, duration, and mode of administration, as well as ethical approvals, statistical approaches, and biomarker evaluations. Additionally, a review of the literature was conducted to determine the influence of phytoconstituents found in *P. dactylifera* on the host system’s oxidative stress-mediated damages [29].

### 2.5. Quality Assessment

The Newcastle–Ottawa scale is a technique used to assess bias risk in cross-sectional studies. This tool covers multiple categories, with precise attention placed on elements like the study outcomes, statistical analysis description, and sample design. The author carried out a blind assessment of the study’s quality, and any disagreements were settled through conversations with an authority on the topic [30]. Articles that are part of the study have to be eligible for it, which is determined by their Newcastle–Ottawa scale score, which cannot be less than 3.

### 2.6. Representation of Data

The population’s data were gathered, documented, examined, and shown as either tabular data or figures. Figure 2 shows the results of the PRISMA search for scientific articles in the literature. Table 1 lists the significant phytoconstituents found in *P. dactylifera* that have been linked to antioxidant capacity. Figure 3 summarizes the potential mechanism for the neuroprotective impact.

## 3. Oxidative Stress in Neurodegeneration

The redox reaction of oxidation involves the loss of electrons [31]. Reactive oxygen species (ROS), which are constantly produced throughout cell metabolism processes, are one byproduct of oxidation [32]. Free radicals from cell catabolism are thought to play a role in aging and other degenerative disorders [33]. Because of this, the human body contains built-in antioxidant defenses against free radicals, including glutathione (GSH), catalase, and superoxide dismutase (SOD) [34]. Oxidative stress is brought on by the build-up of ROS in a compromised antioxidant nerve system [35]. Pro-apoptotic Bcl-2 family members are involved in a neuronal cell death event that is triggered by oxidative stress [36].

The process of lipoperoxidation (LPO) is one way that OS damages lipids [37]. Hydroperoxides, including propanal, hexanal, 4-hydroxynonenal, isoprostane, and malondialdehyde (MDA), are the primary products of LPO [38]. Additionally, when ROS interact with guanine nucleotides, they can cause structural damage to DNA [39]. Under typical circumstances, these metabolites are restored by the oxoguanine glycosylase (hOGG1) enzyme and are collectively referred to as OS biomarkers [40].

Because of the high concentration of polyunsaturated fatty acids (PUFA), such as linoleic acid and arachidonic acid, the human brain is susceptible to oxidative stress [41]. In the neurological system, oxidized PUFA and ROS react to generate lipid peroxidation products, such as lipid peroxyl radical, which starts a chain reaction that further oxidizes PUFA [42]. Therefore, by scavenging peroxyl radicals, an antioxidant system is required to break the chain reaction of free radicals [41,43].

## 4. Major Phytochemicals of *Phoenix dactylifera* with Antioxidant Potential

Perhaps because of worries about the adverse effects of manmade chemicals, there has been a spike in interest in discovering novel antioxidants from plants. Many investigations have been carried out to clarify *P. dactylifera*’s naturally occurring chemicals and indicated the presence of several phenolic compounds [44].

*P. dactylifera* dates contain the following phenolic components: caffeic acid, catechin, and rutin. The ripening stage also affected *P. dactylifera*’s phenolic content, which ranged from 10 mg/100 g to 290 mg/100 g. The polyphenol content of *P. dactylifera* dates was found to be greatest at the kimri stage (290 mg/100 g), followed by the khalal (150 mg/100 g), rutab (20 mg/100 g), and tamr (10 mg/100 g) stages. The most prevalent phenolic compounds discovered in *P. dactylifera* dates were derivatives of gallic acid, ferulic acid, and p-coumaric acid [45,46,47].

Similar to this, it was discovered that *P. dactylifera*’s primary phenolic components and acids at various ripening stages included protocatechuic acid, hydroxybenzoic acid, vanillic acid, gallic acid, isovanillic acid, chlorogenic acid, ferulic acid, isoferulic acid, caffeic acid, hydroxycinnamic acid, and chlorogenic acid. Gallic acid, caffeic acid, chlorogenic acid, syringic acid, p-coumaric acid, m-coumaric acid, and ferulic acid were identified by many researchers as the main phenolics and acids in roasted *P. dactylifera* pits [48,49,50,51,52].

There are several subclasses of flavonoids, including anthocyanins, iso-flavones, flavones, and flavonols, identified in *P. dactylifera*. Active flavonoids, such as luteolin, rutin, isoquercetin, apgenin, and quercetin, are abundant in *P. dactylifera*. Within the kingdom of plants, flavonols are the most prevalent type of flavonoid. Flavonols, also known as oglycosides, are abundant in *P. dactylifera* flesh and pits. The amounts of these substances, however, differed significantly between the *P. dactylifera* fruit’s meat and pits. Chrysoeriol-7-O-(2,6-dirhamnosyl)-glucoside has recently been found in *P. dactylifera* fruit. Using LC-MS/MS techniques, researchers examined the flavonoid composition of *P. dactylifera* at various ripening stages and discovered significant amounts of luteolin, kaempferol, myricetin, naringenin, apigenin, and quercetin [53,54,55,56,57].

Triterpenoids such as lupeol and lup-20(29)-en-3-one, steroids such as b-sitosterol, b-sitosteryl-3-O-b-glucoside, and b-sitosteryl-3b-glucopyranoside60-O-palmitate, and phthalates such as bis(2-ethylheptyl) phthalate and bis(2-ethylhexyl) terephthalate are among the other minor ingredients in *P. dactylifera* fruit. *P. dactylifera*’s bioactive components contribute to its antioxidant and anti-inflammatory qualities. An important quantity of anthocyanidins is also present in *P. dactylifera*, which is primarily found in the kimri stage. A few significant organic acids, such as succinic, oxalic, malic, citric, isobutyric, and formic acids, are present in *P. dactylifera*. These acids enhance *P. dactylifera*’s functionality even more [58,59,60,61,62]. Table 1 displays the chemical structures of the main phytochemicals that are present in *P. dactylifera*.

**Table 1 pharmaceuticals-17-01221-t001:** Important phytoconstituents of *Phoenix dactylifera* identified for antioxidant properties, retrieved from the selected articles.

Sl. No.	Name	Chemical Structure	Reference (s)
1	Caffeic acid	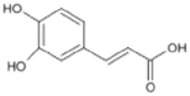	[47,48,49]
2	Ferulic acid	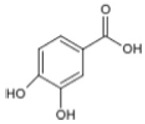	[46,47,48]
3	Catechin	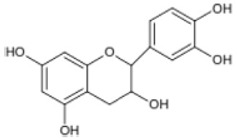	[54]
4	Gallic acid	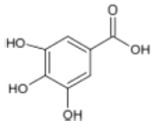	[53,55]
5	p-coumaric acid	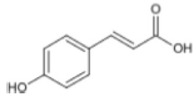	[49,51]
6	Resorcinol acid	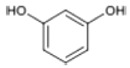	[59,60,61]
7	Quercetin	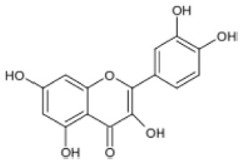	[52,57]
8	Protocatechuic acid	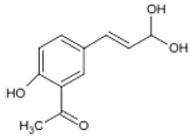	[49,62]
9	Rutin	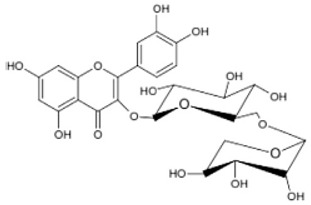	[56,58]
10	Apigenin	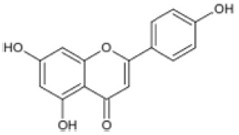	[55,59]

## 5. Antioxidant Properties of *P. dactylifera*

Because *P. dactylifera* fruits contain higher levels of phenolics, melatonin, carotenoids, and vitamins, they are used commonly in Arabian nations and have strong antioxidant potential [63]. Most studies on *P. dactylifera* fruit’s antioxidant potential have used alcoholic and aqueous extracts. Most of the hydrophilic antioxidants in *P. dactylifera* fruit exhibit powerful antioxidant action in the lipid membrane system [64]. When compared to alcoholic extracts, it was found that the aqueous extract of *P. dactylifera* exhibited considerable antioxidant activity. Lipid peroxidation was prevented to an extent of up to 91% using different solvents such as ethyl acetate, methanolic, and aqueous for extraction of *P. dactylifera* in the MTT (3-(4,5-dimethylthiazol-2-yl)-2,5-diphenyltetrazolium bromide) assay [65]. Furthermore, the antioxidant potential of Phoenix dactylifera was found to be comparable to that of well-established herbal remedies such as *Panax notoginseng* and *Ginkgo biloba* [13,14].

In cadmium-intoxicated animals, the ethanolic extract of *P. dactylifera* resulted in a decrease in lipid hydroperoxides and an increase in blood antioxidant enzyme levels [66]. *P. dactylifera may exert* its antioxidant action by suppressing free radicals, which, in turn, slows down the spread of disease [67].

*P. dactylifera* has been shown to have potent antioxidant benefits by additional research. *P. dactylifera* extracts have been shown to stop the depletion of important antioxidants such glutathione peroxidase, superoxide dismutase, and carnitine acyltransferase [68]. Strong antioxidant activity was observed in an investigation using several *P. dactylifera* extracts, with methanolic extracts exhibiting equipotent with gallic acid [69]. Furthermore, they demonstrated potent radical scavenging ability in DPPH and lipid peroxidation tests using *P. dactylifera* date acetone extracts. In contrast to other solvents, they did note the exceptional radical scavenging activity of *P. dactylifera* aqueous extracts [70].

In vivo and in vitro studies on the neuroprotective properties of caffeic acid are reported in the literature. While the hybrid molecule of 20 mg/kg caffeic acid-syringic acid showed excellent neuroprotective effects on transient cerebral ischemia injury in the hippocampal CA1 area, caffeic acid in gerbils established only modest neuroprotection [71]. Due to its potent antioxidant properties, caffeine exhibited dose-dependent protection for neuronal cells against H_2_O_2_-induced cytotoxicity [72]. In a different study, caffeine reduced AChE activity in the cortex and striatum, which enhanced learning and memory in an inhibitory avoidance task [73].

## 6. Putative Mechanisms of Antioxidant Action of *P. dactylifera* in Neurodegenerative Diseases

The precise processes behind *P. dactylifera*’s scavenging action are still unknown, despite the organism’s well-documented antioxidant properties. Based on prior research on the plant extract and its chief phytochemicals, which were covered above, we propose potential pathways for its therapeutic antioxidant actions (Figure 3). In summary, phenolic compounds can respond directly to ROS by scavenging free radicals and demonstrate antioxidant activity by preventing the death of neurons [74].

It is possible that their antioxidative effects were mediated by the polyphenol compounds’ ability to transfer an electron to the free radicals, stabilizing them. The antioxidative activities of phenolic compounds were reported to reduce the Bax/Bcl-2 ratio and caspase-3 production, pro-apoptotic signals, and lipid peroxidation, thus preventing cell death [75]. On the other hand, SOD’s antioxidant enzyme increased [76]. Meanwhile, *P. dactylifera* extract increased the levels of endogenous antioxidant GSH and catalase, which, in turn, decreased lipid peroxidation and demonstrated effects that protected cells [77]. The expression of the neuronal marker genes tyrosine hydroxylase (TH) and brain-derived neurotropic factor (BDNF), which are critical for cell survival because they are involved in neurotransmitter synthesis, also mediates the neuroprotective effects of phenols [78].

Moreover, selenium, which is known to have strong antioxidant properties, is present in *P. dactylifera* [79]. Numerous studies have shown that the main source of this essential trace element’s antioxidant action is selenocysteine residues, which are an essential component of the ROS-detoxifying seleno-enzymes (GPx, thioredoxin reductases, and, maybe, selenoprotein P) [80,81,82]. The combination of selenium and other phenolic compounds seems to be the most plausible source of documented antioxidant and free radical scavenging properties [83].

Rats were given a methanolic extract of *P. dactylifera* fruits at doses of 30, 100, and 300 mg/kg for 15 days as part of a study on the neuroprotective effects of *P. dactylifera* against bilateral common carotid artery blockage. At dosages of 100 and 300 mg/kg, *P. dactylifera* extract significantly reduced the ischemia-induced diminution of GSH, SOD, and CAT expression; however, at lower dosages (30 mg/kg), no significant changes were seen [84]. The methanolic extract of *P. dactylifera* consequently showed brain damage and antioxidant protection that was dose-dependent [85].

In a different study, *P. dactylifera*’s compounds, which include ellagic acid, epicatechin, catechin, kaempferol, quercetin, and apigenin, were shown to have the capacity to function as dual inhibitors of acetylcholinesterase (AChE) and COX-2, potentially improving both cholinergic and inflammatory disorders. In contrast, the compounds cinnamic acid, hesperidin, hesperetin, narengin, and rutin were found to be solely responsible for improving cholinergic transmission. *P. dactylifera* provides neuroprotection against LPS-induced cognitive impairments in rats by preventing neuroinflammation and enhancing cholinergic function [86]. The results showed that extract therapy, probably because of its antioxidant properties, may shield cortical neurons from brain injury [87].

Date seed extract’s potential to prevent cerebral ischemia in male rats was investigated. This study shows that seed extract greatly reduces neuronal damage [84]. The application of seed extracts also retained the ultra-structures of cortical neurons. The group that received seed extract treatment also displayed improvements in fall out latency times [88]. Additionally, the brain’s oxidative stress decreased and its antioxidative enzymes were restored [89]. In addition to these advantages, *P. dactylifera* seed extract also lessens muscle weakness, which protects against the damage caused by ischemia-reperfusion [90]. *P. dactylifera* may have a cerebroprotective effect because of its antioxidant activity. According to this study, using *P. dactylifera* to treat cerebral ischemia may be advantageous [89,91].

The neuroprotective effect of *P. dactylifera* fruits has also been shown by another study, which used male rats in which aluminium chloride was used to induce experimental Alzheimer’s disease. The results of this activity clearly show that *P. dactylifera* is a neuroprotective agent due to its antioxidative qualities, and this activity is attributed to the presence of polyphenolic components such as flavonoids, plant sterols, and ascorbic acid [92]. Furthermore, it has been found that caffeic acid inhibits the synthesis of AChE and the activation of Keap1-Nrf2 [93]. From these data, it is clear that *P. dactylifera* has neuroprotective qualities. This is mainly because the plant contains naturally occurring antioxidants such as phenols, which were previously mentioned.

## *7.* Future Implications

Researchers have placed a lot of emphasis on phenolic chemicals that have been extracted from plants because of their strong antioxidant properties. The specificity in the etiology of neurodegenerative defects may contribute to the ongoing debate about the role of oxidative stress in neurodegeneration [94]. Antioxidant supplementation has been demonstrated to have no advantages in certain clinical trials, and in some cases, it has even had negative effects on the cognitive performance of people with Alzheimer’s disease [95]. Nonetheless, the high oxygen consumption of the brain makes it clear that neuronal cells are extremely susceptible to oxidative stress [96]. Consequently, one of the possible causes of neurodegeneration may be compromised antioxidant defense systems, indicating the significance of antioxidants in halting or postponing the beginning of neurodegeneration [97]. Drugs derived from plants are reported to exhibit a balancing act in the suppression of oxidative stress-induced damages [25]. One of the reasons identified is due to the presence of different classes of phytochemicals that can produce multiple mechanisms [12]. Hence, there is growing interest in determining the potential of plant-based products in overcoming oxidative stress-mediated disorders including neurological defects.

A growing body of evidence indicates that *P. dactylifera*, and especially its polyphenolic components, exhibited potent antioxidant properties. The literature selected for this study identified the presence of vital phytocomponents in *P. dactylifera* that have antioxidant potential. *P. dactylifera*’s main active ingredients are polyphenols, which have strong antioxidant properties [98]. Even though *P. dactylifera* showed strong antioxidant activity, there is still a growing amount of preclinical evidence and no clinical literature to date to adequately support the neuroprotective capabilities of *P. dactylifera*. Although potential pathways for *P. dactylifera*’s antioxidant actions have been suggested, more research is needed to clarify these mechanisms to better understand how *P. dactylifera*’s antioxidant actions relate to its neuroprotective effects.

Alternatively, *P. dactylifera*’s antioxidant benefits may be due to the synergy of several different naturally occurring bioactive compounds in the plant, meaning that rather than products made from a single bioactive compound extracted from the plant, the putative effects would need to be produced by oral administration of the whole plant extract as a dietary supplement [99]. Thus, more preclinical and clinical research examining *P. dactylifera*’s medicinal potential, particularly in the prevention of neurodegeneration, is necessary due to its targeted targeting of neuroprotection through antioxidant activities.

## *8.* Conclusions

According to the study’s research, *P. dactylifera* may have antioxidant characteristics that have neuroprotective effects. By scavenging free radicals and enhancing the antioxidant defense system, the phenolic chemicals found in the plant may be the cause of the oxidative stress attenuation. The fruit is taken for its spiritual and therapeutic benefits and is well-liked throughout the world. Further investigation into the different phytoactive components may reveal a possible naturally derived medication option for the treatment of a range of conditions, including neurodegenerative illnesses.

## Figures and Tables

**Figure 1 pharmaceuticals-17-01221-f001:**
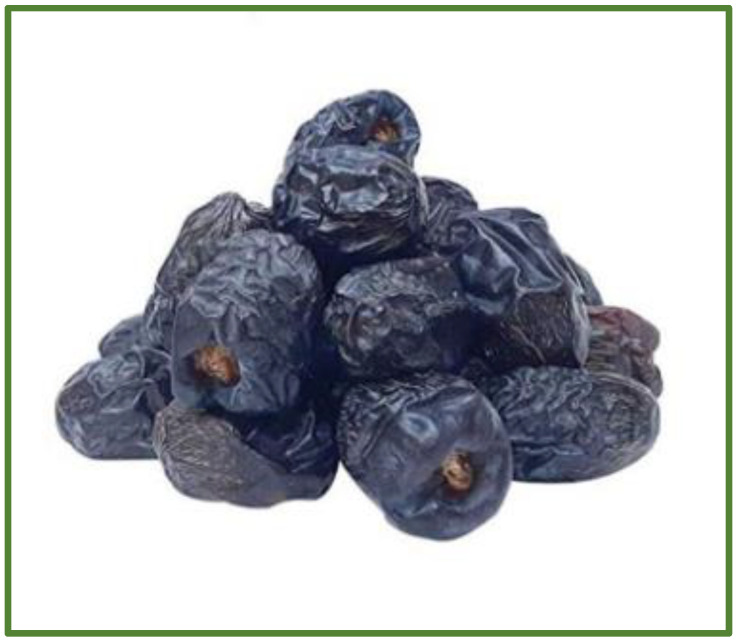
*Phoenix dactylifera* fruits cultivated in Madina province of Saudi Arabia [20,21].

**Figure 2 pharmaceuticals-17-01221-f002:**
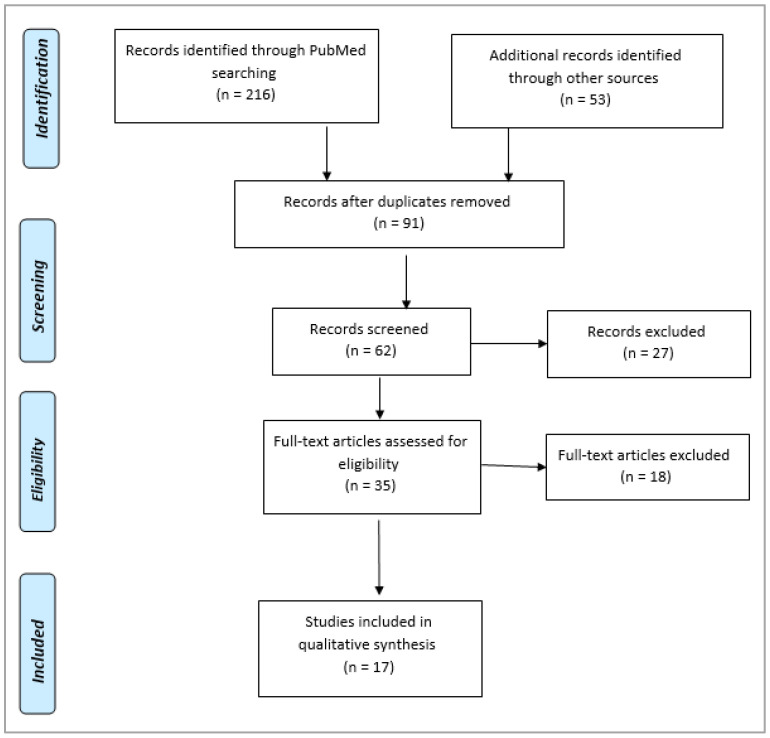
PRISMA flow diagram to select the scientific studies from the literature.

**Figure 3 pharmaceuticals-17-01221-f003:**
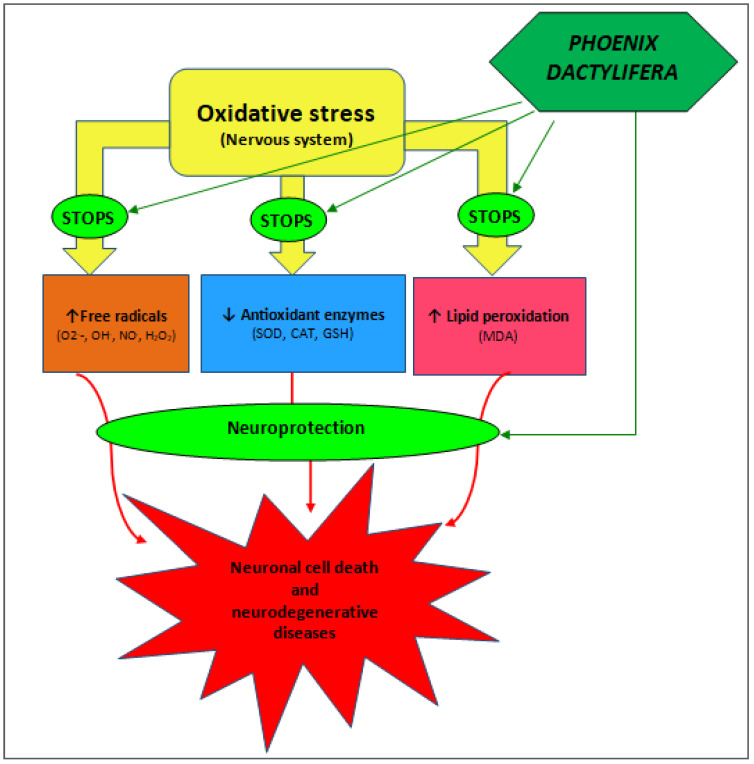
Potential mechanism for the neuroprotective effect of *Phoenix dactylifera*.

## Data Availability

Data are contained within the article or Appendix A.

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
