# Peer review of "Unveiling the Neuroprotective Potential of Date Palm (Phoenix dactylifera): A Systematic Review"

_pharmaceuticals, 2024, doi:10.3390/ph17091221_

Round 1

Reviewer 1 Report

Comments and Suggestions for Authors

The Manuscript entitled " Unveiling the Neuroprotective Potential of Date Palm (Phoenix dactylifera): A Systematic Review has been critically reviewed and can be accepted after minor revision.

The manuscript is interesting and well-written but some comments should be corrected.

Comments

1- Line 76, please write Curcuma longa in the italic.

2- Lines 76-82, delete this paragraph and concentrate on P. dactylifera

3- Lines 106-110, delete this paragraph and concentrate on P. dactylifera

4- Line 275, please change “H2O2-induced” into the subscript form

5- Line 294 “TH and BDNF” Please add the name first then add the abbreviations.

6- Line 316, P.dactylifera should be italics throughout the whole review.

Author Response

Reviewer 1

The Manuscript entitled " Unveiling the Neuroprotective Potential of Date Palm (Phoenix dactylifera): A Systematic Review” has been critically reviewed and can be accepted after minor revision.

The manuscript is interesting and well-written but some comments should be corrected.

Comments

1- Line 76, please write Curcuma longa in the italic.

2- Lines 76-82, delete this paragraph and concentrate on P. dactylifera

3- Lines 106-110, delete this paragraph and concentrate on P. dactylifera

4- Line 275, please change “H2O2-induced” into the subscript form

5- Line 294 “TH and BDNF” Please add the name first then add the abbreviations.

6- Line 316, P.dactylifera should be italics throughout the whole review.

Reviewer-1:

Sl.

No.

Comments

Reply to comments

Line # in the revised manuscript

1

Line 76, please write Curcuma longa in the italic.

As per the suggestion of the honorable reviewer, corrections are made in the revised manuscript.

82

2

Lines 76-82, delete this paragraph and concentrate on P. dactylifera

We agree with the suggestion, the suggested paragraph is deleted.

82-88

3

Lines 106-110, delete this paragraph and concentrate on P. dactylifera

Accordingly, the paragraph is deleted in the revised manuscript.

111-117

4

Line 275, please change “H2O2-induced” into the subscript form

We regret the typo mistake. Correction is done in the revised manuscript.

300

5

Line 294 “TH and BDNF” Please add the name first then add the abbreviations.

In agreement with reviewers’ comment, abbreviations were explained in the revised manuscript.

319,320

6

Line 316, P.dactylifera should be italics throughout the whole review.

We have checked whole manuscript and corrected the name of P. dactylifera.

Whole

manuscript

Reviewer 2 Report

Comments and Suggestions for Authors

Line 18: Give the Family name of the plant.

Abstract: What was the review method? How many studies have been found and used?

Introduction, Lines 36-53: here is nothing new, this is well-known. The introduction is too long and general.

Lines 54-55: "The current medications' efficacy has been reduced due to the growth of drug resistance and tolerance." Which "medications'" the authors mean here? Must be condensed.

Use italics in plant scientific names.

Line 139: "Research that did not satisfy the qualifying requirements was not included." Not clear what the authors want to say here.

Lines 143-144: "Two authors of this work independently reviewed the literature to determine P. dactylifera's involvement in neurodegenerative illnesses." How did these two authors compare their results?

The section "3. Oxidative stress in neurodegeneration" is too long and general, not directly related to the plant.

4.1. Phenolic compounds and 4.2 Flavonoids: flavonoids also belong to phenolic compounds. Line 212: Rutin is also one of the flavonoids but is mentioned under "phenolic compounds".

Lines 225-226: "There are several subclasses of flavonoids, including anthocyanins, isoflavones, flavones, and flavanols." Again, this is well-know fact.

Line 235: "Triterpenoids such as lupeol and lup-20(29)-en-3-one, steroids such..." Triterpenoids are not flavonoids. Here, the authors must use a suitable subtitle.

Line 240: Anthocyanidins must be described in the section of Flavonoids, not after triterpenoids.

Line 256: "(3-(4,5-Dimethylthiazol-2-yl)-2,5-Diphenyltetrazolium Bromide)" No need to use "D" and "B".

5. Antioxidant properties of P. dactylifera: How strong are the antioxidant properties of P. dactylifera if compared with some other plants?

Figure 3: Really "stops" or reduces?

How can the authors characterise the scientific level of studies they describe?

Author Response

Comments and Suggestions for Authors

Line 18: Give the Family name of the plant.

Abstract: What was the review method? How many studies have been found and used?

Introduction, Lines 36-53: here is nothing new, this is well-known. The introduction is too long and general.

Lines 54-55: "The current medications' efficacy has been reduced due to the growth of drug resistance and tolerance." Which "medications'" the authors mean here? Must be condensed.

Use italics in plant scientific names.

Line 139: "Research that did not satisfy the qualifying requirements was not included." Not clear what the authors want to say here.

Lines 143-144: "Two authors of this work independently reviewed the literature to determine P. dactylifera's involvement in neurodegenerative illnesses." How did these two authors compare their results?

The section "3. Oxidative stress in neurodegeneration" is too long and general, not directly related to the plant.

4.1. Phenolic compounds and 4.2 Flavonoids: flavonoids also belong to phenolic compounds. Line 212: Rutin is also one of the flavonoids but is mentioned under "phenolic compounds".

Lines 225-226: "There are several subclasses of flavonoids, including anthocyanins, isoflavones, flavones, and flavanols." Again, this is well-know fact.

Line 235: "Triterpenoids such as lupeol and lup-20(29)-en-3-one, steroids such..." Triterpenoids are not flavonoids. Here, the authors must use a suitable subtitle.

Line 240: Anthocyanidins must be described in the section of Flavonoids, not after triterpenoids.

Line 256: "(3-(4,5-Dimethylthiazol-2-yl)-2,5-Diphenyltetrazolium Bromide)" No need to use "D" and "B".

  1. Antioxidant properties of P. dactylifera: How strong are the antioxidant properties of P. dactylifera if compared with some other plants?

Figure 3: Really "stops" or reduces?

How can the authors characterise the scientific level of studies they describe?

Reviewer-2:

Sl.

No.

Comments

Reply to comments

Line # in the revised manuscript

1

Line 18: Give the Family name of the plant.

According to the suggestion of respected reviewer, addition is done in the revised manuscript.

18

2

Abstract: What was the review method? How many studies have been found and used?

We agree with the queries. Accordingly, we have added the methodology and number of articles screened as well as selected in the present study.

22-24

3

Introduction, Lines 36-53: here is nothing new, this is well-known. The introduction is too long and general.

In agreement with the comment, we have concise the data to indicate only the important information related to the study.

40-57

4

Lines 54-55: "The current medications' efficacy has been reduced due to the growth of drug resistance and tolerance." Which "medications'" the authors mean here? Must be condensed.

Accordingly, we have concise the contents and revised the sentence to indicate clear information.

58-69

5

Use italics in plant scientific names.

We have checked whole manuscript and corrections are done at all necessary places

Whole manuscript

6

Line 139: "Research that did not satisfy the qualifying requirements was not included." Not clear what the authors want to say here.

We have modified the sentence to clarify the information.

148-149

7

Lines 143-144: "Two authors of this work independently reviewed the literature to determine P. dactylifera's involvement in neurodegenerative illnesses." How did these two authors compare their results?

The authors segregated the neurodegenerative diseases and analyzed the role of P.dactylifera on individual diseases. In this way the overlapping and repetition of data was avoided. We have provided this information in the revised manuscript.

154-157

8

The section "3. Oxidative stress in neurodegeneration" is too long and general, not directly related to the plant.

In agreement with reviewers’ comment, we have modified whole section (section-3) and retained only important information that directly relates to the role of oxidative stress in neurodegenerative diseases.

193-224

9

4.1. Phenolic compounds and 4.2 Flavonoids: flavonoids also belong to phenolic compounds. Line 212: Rutin is also one of the flavonoids but is mentioned under "phenolic compounds".

We appreciate the in-depth knowledge of the reviewer. We have corrected the mistake in the revised manuscript.

230,245,247

10

Lines 225-226: "There are several subclasses of flavonoids, including anthocyanins, isoflavones, flavones, and flavanols." Again, this is well-know fact.

We have modified the sentence as ‘There are several subclasses of flavonoids, including anthocyanins, iso-flavones, flavones, and flavonols identified in P. dactylifera’. Hope this will clarify the sentence.

247

11

Line 235: "Triterpenoids such as lupeol and lup-20(29)-en-3-one, steroids such..." Triterpenoids are not flavonoids. Here, the authors must use a suitable subtitle.

Yes, we agree with the comment. Hence to avoid confusion, we have removed the ‘flavonoid’ subtitle.

256-259

12

Line 240: Anthocyanidins must be described in the section of Flavonoids, not after triterpenoids.

In agreement with reviewers’ comment, the contents will be described under ‘Major phytochemicals of Phoenix dactylifera with antioxidant potential’. There is no subtitle in section-4.

260-265

13

Line 256: "(3-(4,5-Dimethylthiazol-2-yl)-2,5-Diphenyltetrazolium Bromide)" No need to use "D" and "B".

Ok. We have corrected these in the revised manuscript.

279

14

5. Antioxidant properties of P. dactylifera: How strong are the antioxidant properties of P. dactylifera if compared with some other plants?

The antioxidant potential of P. dactylifera was found to similar with known herbal medications such as Panax notogiinseng and Ginkgo biloba. We have added this information in the revised manuscript.

279-281

15

Figure 3: Really "stops" or reduces?

We are thankful to reviewer for this suggestion. Yes, it is better to put ‘Reduce’ instead of ‘Stop’. The modified figure-3 is added.

366

16

How can the authors characterise the scientific level of studies they describe?

A comprehensive analysis was done for the data collected from literature. We suggested oxidative stress as one of the markers for neurodegeneration and determined the influence of antioxidants on it. Further, we described the various antioxidant identified as well as isolated from P. dactylifera. A possible mechanism was drawn for the neuroprotection based on the findings from previous research. In the future implications, we have suggested how more research on this plant-based product might aid in identifying the neuroprotective potential that could benefit the population consuming it different regions.

369-403

Reviewer 3 Report

Comments and Suggestions for Authors

Interesting review topic. Some aspects need improvement:

- Methods: In the supplementary file, please provide your electronic search thread used for each database (Section 2.1). 

- Figure 2: The number of articles found in PubMed has been stated. You have conducted the search in multiple databases (PubMed, SCOPUS, Web of Science, and BIOSIS) - there is something wrong here. Please specify the number of identified records in each database separately at the beginning of the flow chart.

 - The position of the figures is incorrect. Please check. You have cited them in Section 2.6. Section 2.6 is not needed. 

- Create a table and outline the findings of the "17" included studies. 

- I don't see the significance of adding table 1. It can be removed. They don't add much to the review.

- In section 7, clearly list the identified gaps in those 17 studies, providing clear (points) for future research.

Comments on the Quality of English Language

Fine - general proof reading needed.

Author Response

Reviewer 3

Comments and Suggestions for Authors

Interesting review topic. Some aspects need improvement:

- Methods: In the supplementary file, please provide your electronic search thread used for each database (Section 2.1). 

- Figure 2: The number of articles found in PubMed has been stated. You have conducted the search in multiple databases (PubMed, SCOPUS, Web of Science, and BIOSIS) - there is something wrong here. Please specify the number of identified records in each database separately at the beginning of the flow chart.

 - The position of the figures is incorrect. Please check. You have cited them in Section 2.6. Section 2.6 is not needed. 

- Create a table and outline the findings of the "17" included studies. 

- I don't see the significance of adding table 1. It can be removed. They don't add much to the review.

- In section 7, clearly list the identified gaps in those 17 studies, providing clear (points) for future research.

Comments on the Quality of English Language: Fine - general proof reading needed.

Reviewer-3:

Sl.

No.

Comments

Reply to comments

Line # in the revised manuscript

1

- Methods: In the supplementary file, please provide your electronic search thread used for each database (Section 2.1).

As per suggestion of the honorable reviewer, we have provided the electronic search thread used for each database in form of supplementary file.

Supplementary file attached

2

- Figure 2: The number of articles found in PubMed has been stated. You have conducted the search in multiple databases (PubMed, SCOPUS, Web of Science, and BIOSIS) - there is something wrong here. Please specify the number of identified records in each database separately at the beginning of the flow chart.

Accordingly, we have included the number of articles retrieved from literature for different databases such as PubMed (n=216), SCOPUS (n=26), Web of Science (n=15), and BIOSIS (n=12).

132-133

3

 - The position of the figures is incorrect. Please check. You have cited them in Section 2.6. Section 2.6 is not needed.

In agreement with the reviewer’s comment, we have shifted the figure-2 to section 2.5.

181

4

- Create a table and outline the findings of the "17" included studies. 

The significant findings from the 17 selected articles is represented in table-1. We have modified the title of the table to clarify the information.

268

5

- I don't see the significance of adding table 1. It can be removed. They don't add much to the review.

The table-1 summarizes the important phytoconstituents of P. dactylifera retrieved from selected 17 articles used in the present study. These phytochemicals have been reported to exhibit antioxidant property.

266-268

6

- In section 7, clearly list the identified gaps in those 17 studies, providing clear (points) for future research.

Accordingly to the suggestion, we have revised and modified the section 7 to clearly indicate the gap and how this study might open the arena for future research.

380-385

387-389

7

Comments on the Quality of English Language: Fine - general proof reading needed.

The whole manuscript is revised and a special care was taken to fix all grammatical errors.

Whole

manuscript

Round 2

Reviewer 2 Report

Comments and Suggestions for Authors

After modifications, I can suggest accepting the manuscript for publication in the journal Pharmaceuticals.

Reviewer 3 Report

Comments and Suggestions for Authors

The authors have made the necessary revisions.